# Cortical propagation tracks functional recovery after stroke

**Gloria Cecchini**[1,2,3]☯*, **Alessandro Scaglione**[2,4]☯, **Anna Letizia Allegra Mascaro**[4,5]☯, **Curzio Checcucci**[2,4], **Emilia Conti**[2,4,5], **Ihusan Adam**[2,3,6], **Duccio Fanelli**[2,3,7], **Roberto Livi**[2,3,7], **Francesco Saverio Pavone**[2,4,8], **Thomas Kreuz**[9]

**1** Department of Mathematics and Computer Science, University of Barcelona, Barcelona, Spain, **2** Department of Physics and Astronomy, University of Florence, Sesto Fiorentino, Italy, **3** CSDC, University of Florence, Sesto Fiorentino, Italy, **4** European Laboratory for Non-linear Spectroscopy, University of Florence, Sesto Fiorentino, Italy, **5** Neuroscience Institute, National Research Council, Pisa, Italy, **6** Department of Information Engineering, University of Florence, Sesto Fiorentino, Italy, **7** INFN, Florence Section, Sesto Fiorentino, Italy, **8** National Institute of Optics (INO), National Research Council (CNR), Sesto Fiorentino, Italy, **9** Institute for Complex Systems (ISC), National Research Council (CNR), Sesto Fiorentino, Italy

☯ These authors contributed equally to this work.
* gloria.cecchini@ub.edu

**Data Availability Statement:** All data generated or analysed during this study have been deposited in https://data.mendeley.com/datasets/gpsjkbp6h4/1.

**Funding:** This project has received funding from the H2020 EXCELLENT SCIENCE - European

## Abstract

Stroke is a debilitating condition affecting millions of people worldwide. The development of improved rehabilitation therapies rests on finding biomarkers suitable for tracking functional damage and recovery. To achieve this goal, we perform a spatiotemporal analysis of cortical activity obtained by wide-field calcium images in mice before and after stroke. We compare spontaneous recovery with three different post-stroke rehabilitation paradigms, motor training alone, pharmacological contralesional inactivation and both combined. We identify three novel indicators that are able to track how movement-evoked global activation patterns are impaired by stroke and evolve during rehabilitation: the duration, the smoothness, and the angle of individual propagation events. Results show that, compared to pre-stroke conditions, propagation of cortical activity in the subacute phase right after stroke is slowed down and more irregular. When comparing rehabilitation paradigms, we find that mice treated with both motor training and pharmacological intervention, the only group associated with generalized recovery, manifest new propagation patterns, that are even faster and smoother than before the stroke. In conclusion, our new spatiotemporal propagation indicators could represent promising biomarkers that are able to uncover neural correlates not only of motor deficits caused by stroke but also of functional recovery during rehabilitation. In turn, these insights could pave the way towards more targeted post-stroke therapies.

## Author summary

Millions of people worldwide suffer from long-lasting motor deficits caused by stroke. Very recently, the two basic therapeutic approaches, motor training and pharmacological intervention, have been combined in order to achieve a more efficient functional recovery.

Research Council (ERC) under grant agreement ID n. 692943 BrainBIT and from the European Union's Horizon 2020 Research and Innovation Programme under Grant Agreement No. 785907 (HBP SGA2) [Grant recipient: F.S.P.]. This research was supported by the EBRAINS research infrastructure, funded from the European Union's Horizon 2020 Framework Programme for Research and Innovation under the Specific Grant Agreement No. 945539 (Human Brain Project SGA3) [Grant recipient: F.S.P.]. The funders had no role in the study design, data collection and analysis, decision to publish, or preparation of the manuscript.

**Competing interests:** The authors have declared that no competing interests exist.

In this study, we analyze the neurophysiological activity in the brain of mice observed with in vivo calcium imaging before and after the induction of a stroke. We use a newly developed universal approach based on the temporal sequence of local activation in different brain regions to quantify three properties of global propagation patterns: duration, smoothness and angle. These innovative spatiotemporal propagation indicators allow us to track damage and functional recovery following stroke and to quantify the relative success of motor training, pharmacological inactivation, and a combination of both, compared to spontaneous recovery. We show that all three treatments reverse the alterations observed during the subacute phase right after stroke. We also find that combining motor training and pharmacological intervention does not restore pre-stroke features but rather leads to the emergence of new propagation patterns that, surprisingly, are even faster and smoother than the pre-stroke patterns.

## Introduction

Stroke is a severe disease that alters cortical processing producing long lasting motor or cognitive deficits. Treatments generally include motor rehabilitation, pharmacological therapies, brain stimulation, or combinations of them [1]. However, the functional outcome measured as behavioral recovery, depends on multiple factors such as age, lesion size and type, edema formation or inflammation [2, 3]. One way to track recovery after stroke is by monitoring cortical activity, which is known to undergo drastic changes that have been tightly linked to structural alterations [4–6]. Previous studies have reported that stroke produces global widespread alterations in cortical activity as measured by changes in resting state functional connectivity or cortical excitability. On the one hand, electrophysiological studies have shown that stroke and recovery modulate both resting state and stimulus evoked cortical oscillations in motor areas [7, 8]. On the other hand, neuroimaging studies have shown that stroke alters the resting state functional connectivity, for example it reduces the interhemispheric correlations between motor networks, and these changes correlate with behavioral deficits [9, 10]. Furthermore, these changes in resting state functional connectivity can be used to discriminate subjects with behavioral deficits [11, 12]. This supports the idea that monitoring how cortical activity evolves over time could be used to track recovery after stroke and it could represent a powerful tool to evaluate the efficacy of stroke treatments or better yet could lead to biomarkers of functional recovery.

Advancements in neural imaging combined with genetically encoded calcium indicators allow to monitor neural activity over almost the entire cortical mantle with high spatial resolution on a sub-second temporal scale [13–18]. Using these tools it has been shown that the cortical activity of animals engaged in a behavioral task is characterized by cortex-wide global activation patterns [19] and these patterns are shaped by learning across sessions [20]. In the context of stroke calcium imaging has been used to demonstrate that stroke profoundly affects resting state functional connectivity [21] and leads to sensorimotor remapping of the peri-infarct area in mice [14, 22]. Moreover, in a recent study, we have shown that movement-related activation maps are different for stroke therapy associated with functional recovery [6]. Therefore neural activity as measured by calcium imaging could be used as an indicator of cortical remapping and of redistribution of functional connectivity among spared regions and could also be directly associated with behavioral recovery.

Here, we propose that damage and functional recovery can be tracked by monitoring the spatiotemporal properties of movement-evoked widespread activation patterns or global

events. In particular, we use our recently proposed SPIKE-order analysis [23] on in-vivo calcium imaging data from the cortex of awake mice to identify global events and to sort the participating regions from first to last (or leader to follower). Additionally, in order to characterize the spatiotemporal properties of each individual global event, we extend this method and define three propagation indicators: duration, smoothness (i.e. how ordered and consistent is the direction of the propagation), and angle.

First, we provide a characterization of global events in healthy controls. We show that these events are mostly associated with the exertion of force and that their duration and direction are modulated by different behavioral events. Then, to understand the impact of stroke on the global events we quantify the three propagation indicators in a group of subacute stroke subjects. We provide evidence that subacute stroke alters the propagation patterns by increasing the duration while decreasing the smoothness of the global events. To test if global events propagation patterns can be used to track recovery and to identify treatments that lead to generalized recovery, we quantify the propagation indicators in a group of subjects with untreated stroke and in three groups of subjects with different therapeutic interventions. The first therapy group received motor training alone which leads to task-specific improvement of the motor functions [24]. The second group was characterized by a transient pharmacological inactivation of the controlesional hemisphere without any improvement in the motor functions. Finally, the third therapy combined both motor training and pharmacological inactivation producing a generalized recovery of the forelimb functionality. While all treatments reduce the effect of stroke on the propagation indicators, the combined rehabilitative therapy leads to new propagation patterns defined by the shortest duration and the highest smoothness among all experimental groups.

## Methods

### Ethics statement

All experimental procedures were performed in accordance with directive 2010/63/EU on the protection of animals used for scientific purposes and approved by the Italian Minister of Health, authorization n.183/2016-PR.

### Experimental set-up and data collection

In this section we provide a short overview of the data and the methods we used to analyze them. For more technical details please refer to the section "Materials and Methods" in the Supplementary Materials.

The aim of this study was to investigate changing propagation patterns during motor recovery from functional deficits caused by the induction of a focal stroke via a photothrombotic lesion. For this purpose, we analyzed calcium imaging signals recorded from 26 mice to compare three different rehabilitative therapies, one that used motor training alone (robot group), one based on a transient pharmacological inactivation of contralesional activity (toxin group) and one that performed both of these two treatments together (combined group).

A schematic representation of the robotic system, the M-Platform [25, 26], is shown in Fig 1A. This system uses passively actuated contralesional forelimb extension on a slide to trigger active retraction movements that were subsequently rewarded (up to 15 cycles per recording session). The effect of the motor activity was monitored via the discrete status of the slide and by recording the force the mice applied to the slide. As a measure of the neural activity itself we performed wide-field calcium imaging over the affected hemisphere, from the somatosensory to the visual areas. Selecting a region of interest of 2.16 x 3.78 mm and spatially

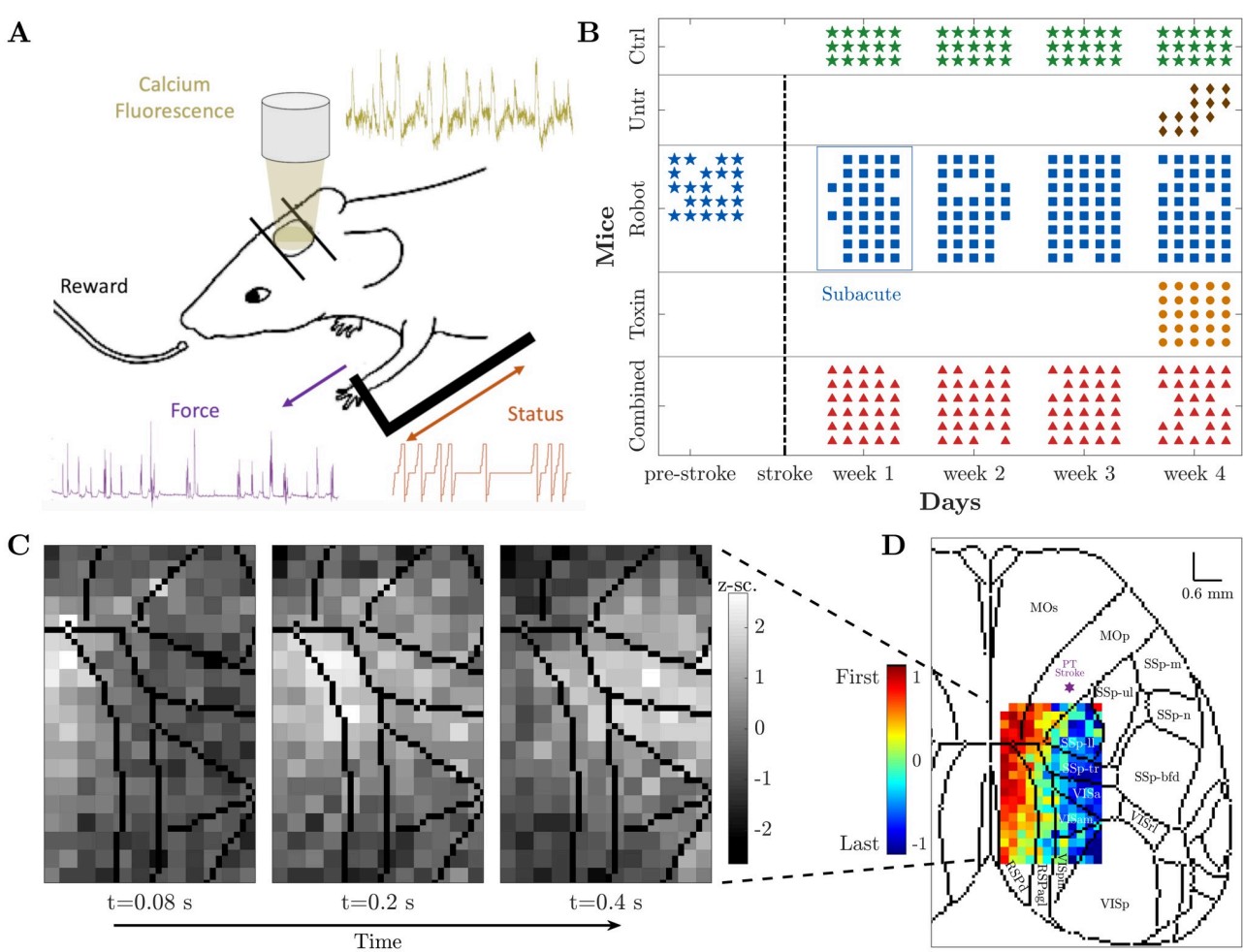

**Fig 1. Experimental set-up and data collection. (A)** Motor-training of mice was performed on the M-Platform, which uses a movable robotic slide for a retraction movement of the left forelimb. Motor activity was monitored via the discrete status of the slide (orange) and the force applied by the mouse to the slide (purple). Meanwhile, the cortical activity (yellow) was recorded using wide-field calcium imaging. **(B)** Recording schedule for each mouse per day. The control (green) and combined treatment (red) group performed four weeks of daily training; for the robot (blue) group we additionally recorded one week before stroke induction (typically five sessions per week). Untreated (brown) and toxin (orange) groups only performed one week of daily training starting four weeks after the lesion. Star symbols refer to the healthy condition. Days of recording move along the horizontal axis, while the vertical axis corresponds to mice divided per group. **(C)** Calcium imaging sequence of cortical activation, superimposed on contours of brain regions according to the standard atlas. **(D)** Propagation pattern, from leader (red) to follower (blue), of the event depicted in (C). Color coding is based on the SPIKE-order (for details see section "SPIKE-order" in "Materials and Methods" (Supplementary Materials)). For a complete list of brain regions and their acronyms see S1 Fig.

downsampling by a factor 3 resulted in calcium images of 12 x 21 pixels of size 180 $\mu$m. These are the signals that we analyzed.

The 26 mice were divided into five groups: control (3 mice), untreated (4 mice), robot (8 mice), toxin (5 mice), and combined treatment (6 mice). The healthy controls had no stroke induced but underwent four weeks of motor training on the M-Platform. The untreated mice performed one week of motor training starting 25 days after injury. The toxin group, which received Botulinum Neurotoxin E (BoNT/E) injection on the contralesional hemisphere right after the photothrombotic damage, also performed one week of motor training starting 25 days after injury. Both robot and combined treatment mice underwent physical rehabilitation on the M-platform for four weeks starting five days after injury. In addition, 5 out of 8 robot mice were also recorded for one week before stroke (5 sessions). For toxin and combined

treatment mice the pharmacological inactivation of the primary motor cortex in the contrale-
sional hemisphere was carried out in order to counterbalance the hyperexcitability of the
healthy hemisphere [27].

The recording schedule for all five groups is shown in Fig 1B. Apart from the data acquisi-
tion during the shared training regime, 5 out of 8 robot mice were also recorded for one week
before the stroke (pre-stroke condition). We have confirmed that this pre-stroke condition
shows no statistical difference, both qualitatively and quantitatively, with the first week of
recordings of the control group, as could be expected for two groups of healthy mice (for
details see S2 Fig).

Fig 1C displays a sequence of snapshots of the calcium activity over time. The three images
illustrate one pull of the slide by the contralateral forelimb of a control mouse, from the activa-
tion of the average calcium activity (left) via its maximum (middle) to its tail end (right). The
Supplementary Materials contains a movie (S1 Video) that covers all 36 frames for this one
training cycle.

The central method of this study was a mapping of this sequence of snapshots into the
propagation pattern shown in Fig 1D. In this matrix the order of activation of the individual
pixels was color-coded from red (earliest) to blue (latest). We superimposed this order matrix
on the standard atlas of brain regions [28] which illustrates that the recording area covers the
primary motor area M1 (the location of the lesion), the primary somatosensory area, and the
primary visual cortex, as well as the retrosplenial cortex.

## Event identification, propagation analysis, and definition of three propagation indicators: Duration, angle, and smoothness

Here we explain our use of the SPIKE-order framework [23] to identify global events and
assess the propagation of activation within these events (compare Fig 1D). In particular, we
focus again on one individual global event to illustrate how we characterize the detailed activa-
tion patterns with three propagation indicators: **duration**, **angle**, and **smoothness**.

In Fig 2 we show the activity during a complete recording session of one mouse. Fig 2A
depicts the status, a discrete codification of the current phase of the passive extension and
active retraction cycle, e.g. position of the slide and acoustic go and reward cues. Most relevant
here are the marked times of the pull completions which typically correspond to peaks in the
force applied to the slide (Fig 2B, force peaks are marked at threshold crossings) but already a
quick look at the numbers shows that the mapping is not perfect. Indeed there are typically
more force peaks than rewarded pull completions (20 force peaks versus 13 status events in
this case). The same peaks, and some more, are also present in the calcium signal computed by
averaging the fluorescence signal over all pixels (Fig 2C, here 23 calcium peaks are marked).

In the next step we looked at all individual pixels but here we do not show all the traces
with their peaks but just small time markers denoting the time of their threshold crossings (Fig
2D). This is very similar to a rasterplot showing in each row the spike train of one individual
neuron and accordingly we here follow this terminology and call the threshold crossings of
individual pixels 'spikes'. The first thing to notice is that while there are a few spikes in the
background (in black), by far most of the spikes are part of global events (in color) matching
the peaks in the average calcium activity shown right above. To automatically identify these
global events and sort the spikes within these events from leader to follower we used the
SPIKE-order framework proposed in [23].

After some initial denoising we first used an adaptive coincidence detector [29] to pair
spikes such that each spike is matched with at most one spike in each of the other pixels. By
means of the symmetric and multivariate measure SPIKE-Synchronization [30] we filtered out

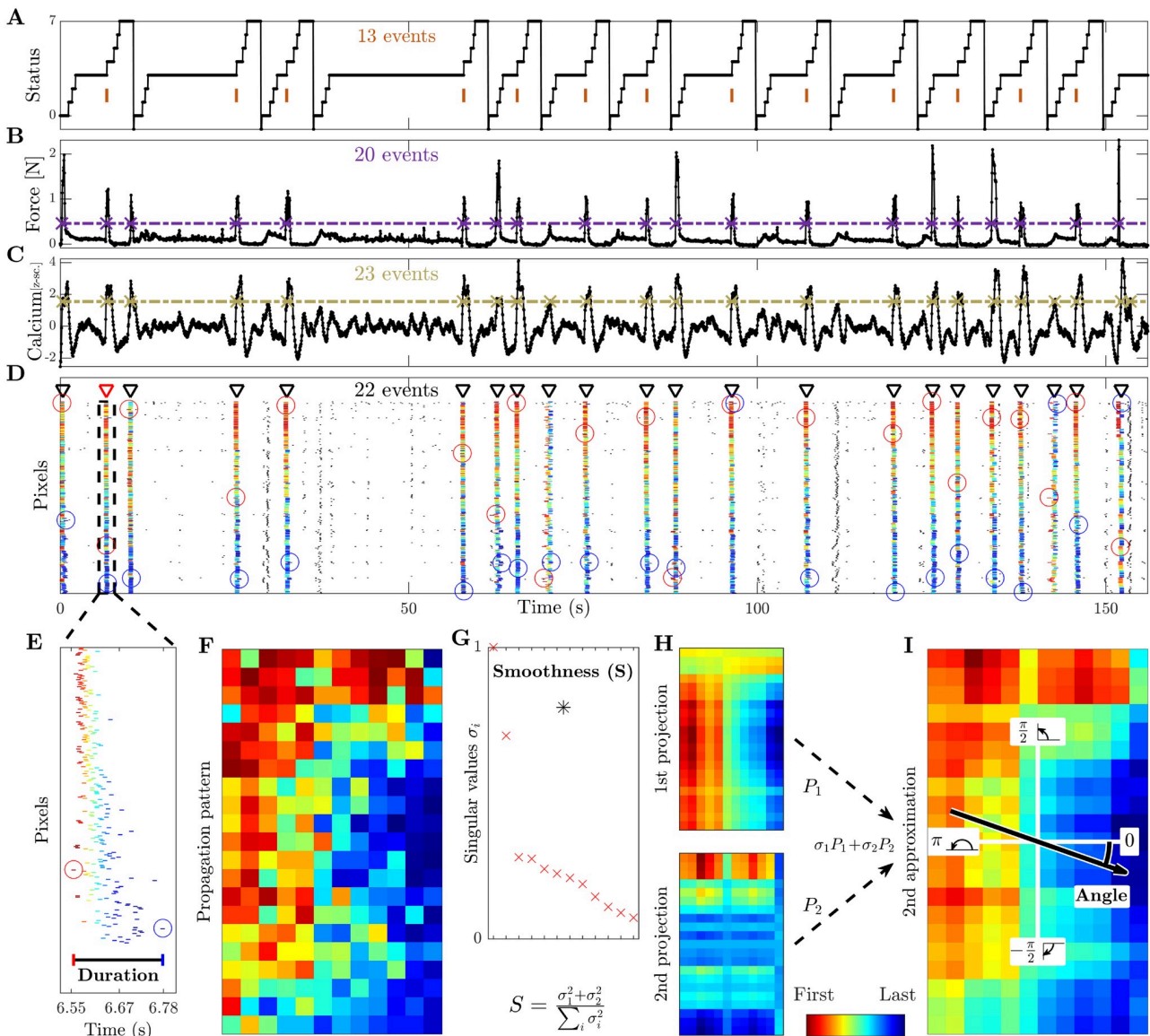

**Fig 2. Event identification, propagation analysis, and definition of three propagation indicators: Duration, smoothness, and angle. (A)** Status of the robotic slide. The longer horizontal plateaus at status 3 correspond to the time interval during which the mouse is allowed to retract the slide. The orange bars refer to the time when the pull is completed and the mouse receives its reward. **(B)** Force applied by the mouse during the retraction movement. **(C)** Average calcium signal over all pixels. The purple and yellow dashed lines below refer to the threshold used to identify the force peaks and the global calcium events, respectively. **(D)** Raster plot obtained from the threshold crossings of individual pixels versus time. Triangle indicators show the global events identified during this session, and the red triangle with dashed box marks the event analyzed in the remaining panels below. The first (last) spike of each global event is marked by a red (blue) circle. Within each subplot (a)-(d) we state the number of the respective events / threshold crossings. **(E)** Zoom of this selected event. The **duration** is defined as the interval from the first to the last spike within the event. **(F)** The propagation matrix is obtained by projecting the relative order of these threshold crossings onto the 2D-recording plane. **(G)** Singular values (red) vs. order of approximation, obtained by means of singular value decomposition (SVD). The **smoothness** (black asterisk) measures the quality of the second order approximation. **(H)** First and second approximations of the propagation matrix and **(I)** second order approximation as their weighted sum (cumulative). The **angle** of the propagation, defined relative to the horizontal axis.

all spikes which were not coincident with spikes in at least three quarters of the other spike trains. To the global events that remained we applied the asymmetric SPIKE-order indicator [23] which quantifies the leader-follower relationship between pairs of spikes. For each event the SPIKE-order is then color-coded from leader (red) to follower (blue). Finally, we used the

scalar Synfire Indicator [23] to also sort the spike trains in the rasterplot from overall leader to overall follower. Since the sorting takes into account all global events, the first spike trains contain more leading spikes (red) and the last spike trains more trailing spikes (blue).

In Fig 2E we zoom in on the fourth global event of the rasterplot. Here we define the first propagation indicator, the event **duration**, as the time from the first to the last spike of this event. The propagation matrix of this specific event, obtained by projecting the color-coded relative order of the spikes onto the pixels of the 2D-recording plane, is shown in Fig 2F. We then used singular value decomposition (SVD) to calculate the two remaining propagation indicators.

SVD [31] searches for spatial patterns by decomposing the propagation matrix into three simple transformations: a rotation, a scaling along the rotated coordinate axes and a second rotation. The scaling is a diagonal matrix which contains along its diagonal the singular values of the propagation matrix. In Fig 2G we depict the (sorted) singular values $\sigma_i$ (rescaled to the highest value). We also show the value of the second propagation indicator, the **smoothness** $S$, which is defined as the relative weight of the first two projections, their sum divided by the sum of all singular values. Backprojecting the (sorted) singular values one at a time resulted in various orders of approximations for the original propagation matrix. The first two such projections are displayed in Fig 2H and the second order approximation, their weighted sum, is shown in Fig 2I. From the weighted average of the mean gradients of these first two projections we calculated the third propagation indicator, the **angle** of the main propagation direction. Note that the smoothness quantifies how well the approximation using only the first two singular values captures the full spatiotemporal pattern obtained by considering all singular values. This can be verified visually by comparing the second approximation (Fig 2I) with the original propagation matrix (Fig 2F).

A comparison of Fig 2D and 2C clearly shows that all global events in the rasterplot can easily be matched with a peak in the average calcium trace, in fact, these are basically two equivalent ways to visualize a peak of global calcium activity. However, while the vast majority of global events are in close proximity of a peak in the force, not all of them are. This we can use to categorize the global events into two groups: Force (**F**) and non-Force (**nF**). Among the force events we can distinguish two kinds of events, a few of them occur during the passive extension of the arm by the slide (Passive, **Pass**) but most of them do not, i.e., they occur during the active retraction phase (Active, **Act**). Finally, among those active events we can differentiate between movements which lead to a completion of the forelimb retraction and therefore are rewarded (Reward Pulling, **RP**) and movements which are not completed and thus not rewarded (non-Reward Pulling, **nRP**). The reward pulling events are the ones that correspond to the vertical markers in the status trace of Fig 2A.

The overall categorization can be visualized by means of this branching structure:

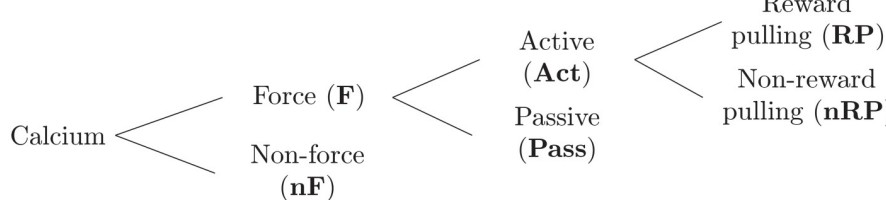

## Results

In this study, we sought a biomarker for functional recovery after stroke in the neural activity propagation. To this aim, we used global event propagation analysis to investigate the spatio-temporal features of neuronal activation over the dorsal cortex. We compared the propagation patterns of healthy mice versus subacute stroke and of untreated versus treated mice. Here we looked at three different treatments (robot, toxin, and combined, see Fig 1B). We decided to investigate three main indicators (duration, smoothness, and angle of the propagation) to account for both temporal and spatial propagation characteristics (compare Fig 2). To identify common brain dynamics associated with similar behaviors, we dealt with each type of event separately.

### Cortical propagation features discriminate event types in healthy mice

We first wondered if neuronal activation in awake healthy mice involved a large region of the cortex and if these global events were related to specific classes of behavioral events in our experimental paradigm. The results in Fig 3 focus on two of the three indicators introduced in Fig 2 and they refer to one healthy mouse during all sessions (four weeks, five days per week, see Fig 1B). Three different example patterns with increasing smoothness and varying angle are depicted in Fig 3A–3C. For low smoothness values, the identified propagation pattern looks random, thus not displaying a clear directionality (Fig 3A), and therefore measures for angle and duration are less meaningful in those cases. On the other hand, high smoothness corresponds to clear patterns (Fig 3B and 3C show two cases of high smoothness but orthogonal directionality). Fig 3D–3F show scatter plots of smoothness against angle for different event types, together with the marginal histograms. While narrowing down the type of event does not reduce the whole range of values, the marginal distributions of the angle and smoothness converge to a peak distribution for the angle centered in 0.46, and to a distribution with mean 0.68 for the smoothness.

Next, we performed a detailed quantitative analysis of all three indicators (duration, smoothness and angle) for all three healthy mice along four weeks of motor training on the M-Platform (control group, Fig 4). The dependency of the three indicators was evaluated both over time and with respect to different event types. In the beginning we analyzed if the occurrence of the global events was associated with the application of forces or if it was unrelated. For the four weeks of recordings in healthy mice the variation of the number of events divided by type is depicted in Fig 4A. Most of these events occurred when the mouse applied a force to the handle (1568, corresponding to 87% of the total number of events). Furthermore, most of the force events (1147) occurred when the mouse was actively pulling the handle (73% of force events, 64% of the total), and 779 of those corresponded to reward pulling events (68% of active events, 43% of the total).

Global events last 0.59 ± 0.04 s with a smoothness of 0.63 ± 0.03, and they propagate along angles −0.3 ± 0.4, see subplot "All" of Fig 4B–4D. The little variation in duration (Fig 4B), smoothness (Fig 4C) and angle (Fig 4D) implies high coherence of the parameters of spatio-temporal propagation over weeks. This suggests that longitudinal motor training in healthy mice does not alter the number of events or the propagation patterns.

Results for duration, smoothness and angle were then analyzed looking at specific event types. Fig 4E shows the event duration for the three consecutive subdivisions of the event types. Further specifying the type of event leads to shorter and shorter propagations with the shortest average duration being obtained for reward pulling events. The same argument can be made for the smoothness (Fig 4F); reward pulling events display the highest smoothness, on average, among all the other events. Fig 4G shows the distributions of the angle; the

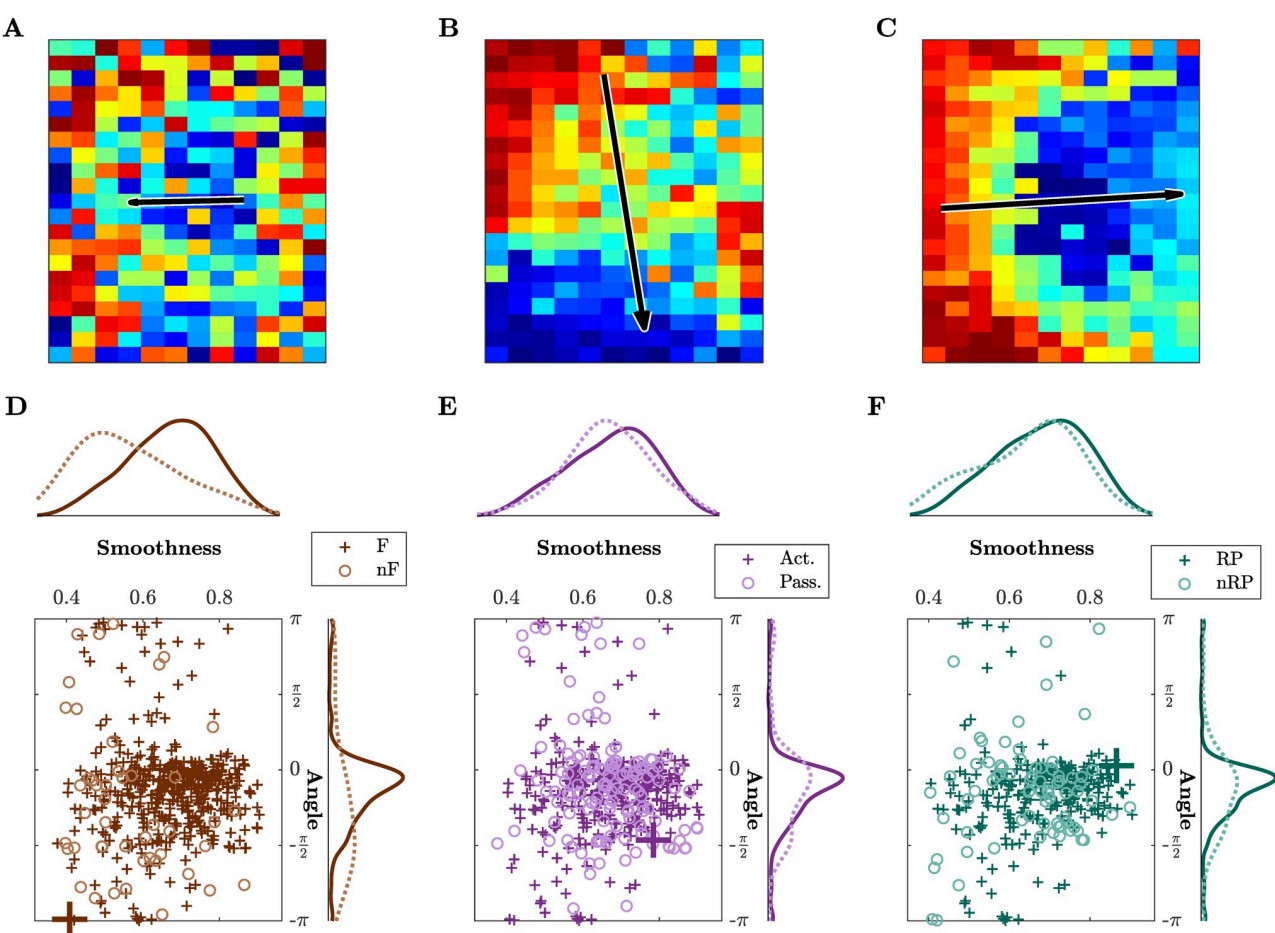

**Fig 3. Propagation pattern varies across event types in healthy mice. (A-C)** Three different patterns with increasing smoothness and varying angle. Note that the length of the arrow is proportional to the smoothness. **(A)** Low smoothness identifies a poor propagation pattern without any proper directionality. **(B)** High smoothness identifies a clear pattern and corresponding directionality. **(C)** High smoothness and horizontal propagation in contrast with (A) low smoothness and opposite horizontal direction, and different from (B) which has still high smoothness, but an almost orthogonal angle of $-\pi/2$. **(D-F)** Scatter plots and histograms of smoothness and angle of propagation for all events. The events whose propagation patterns are shown in (A)-(C) are highlighted by larger markers.—Brown colors refer to force (F) and non-force (nF) events, active (Act) and passive (Pass) events are in purple, reward pulling (RP) and non-reward pulling (nRP) events in green. All plots include the events from all the sessions for one healthy mouse.

propagation becomes more directed when narrowing down the type of event. The difference between event types in the angle distribution reduces as well. Interestingly, also the bimodal nature of the distribution is attenuated. Specifically, the peak at $-\pi/2$ (see first plot in Fig 4G) is initially caused by non-force events, then within force events it stands out in the passive cases, finally it is predominant in the non-reward pulling events. This suggests that task-specific events, such as reward pulling, are characterized by consistent propagation patterns.

The patterns observed in healthy mice are characterized by different spatiotemporal features when comparing force and non-force events in terms of smoothness (p = 0.001) and angle of propagation (p = $10^{-10}$), and when comparing active and passive events in terms of duration (p = 0.002) and angle (p = $10^{-9}$). Also between rewarded and non-rewarded pulls events the observed patterns display different characteristics when comparing smoothness (p = 0.001) and angle (p = $10^{-8}$), see Fig 4E–4G and S1 Table for the p-values.

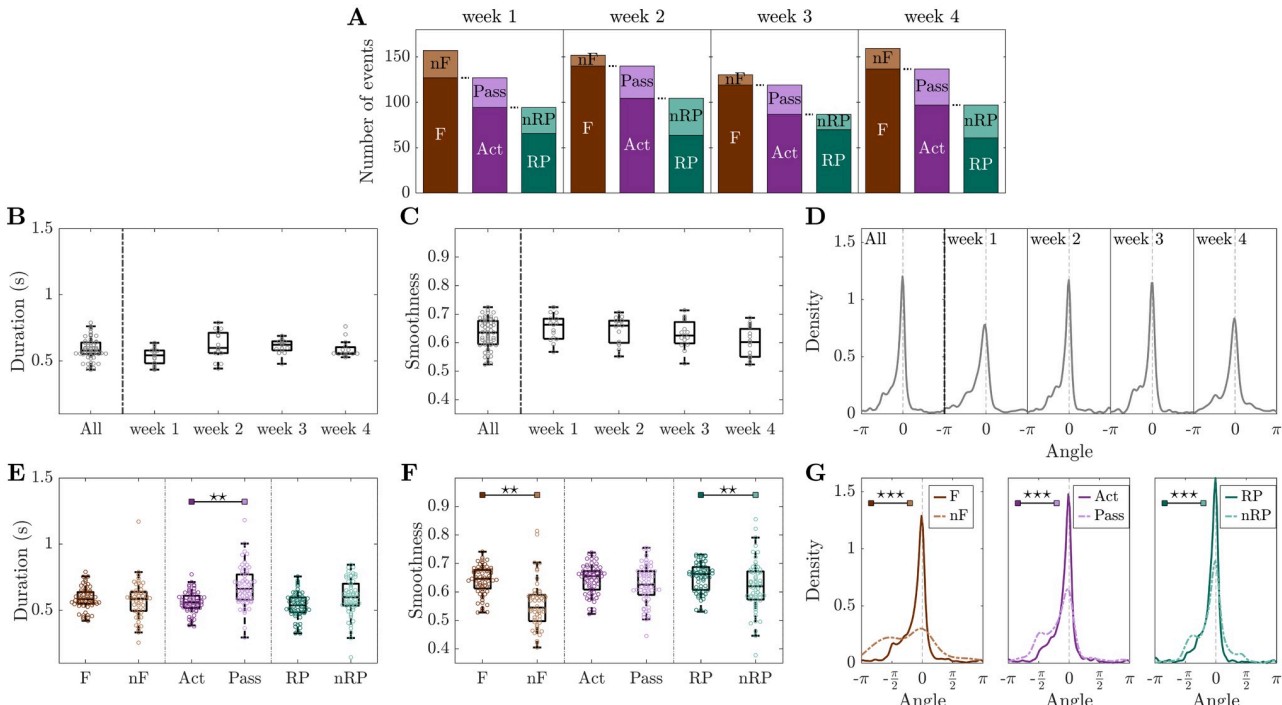

**Fig 4. Cortical propagation features discriminate event types in healthy mice. (A)** Mean number of events per mouse per week, partitioned by type of event. **(B-D)** Duration, smoothness, and angles distribution are preserved over weeks. **(E)** Event duration is a marker for discriminating active (Act) and passive (Pass) events. **(F)** Smoothness discriminates force (F) and non-force (nF) events, as well as reward pulling (RP) and non-reward pulling (nRP) events. **(G)** Narrowing down the type of event leads to more directed propagation patterns.—Duration and angle are weighted by smoothness. Markers in (B,C,E,F) refer to the average value per day. In (B,C) each point is the average of all events per day per mouse, while in (E,F) each point is the average of all events of a specific type per day per mouse. Within each box in (B-F), the central mark indicates the median, and the bottom and top edges of the box indicate the $25^{th}$ and $75^{th}$ percentiles, respectively. Control group n = 3 mice. P-values of statistical tests in S1 Table, "⋆" refers to difference in variance.

In summary, in healthy mice there is a high coherence of the parameters of spatiotemporal propagation over weeks, suggesting that a simple motor task alone does not change the duration, smoothness, and angle of events. Moreover, the investigated spatiotemporal propagation indicators discriminate between different event types when a specific characteristic is taken into account, i.e., force versus non-force, active versus passive, reward versus non-reward pulling.

## Subacute phase after stroke is characterized by an increase of event duration

We pondered how the spatiotemporal propagation indicators were altered by cortical injury, and thus looked at the cortical activation events as associated to classes of behavioral events in the first week after stroke (called subacute stroke). Moreover, we compared these results with the first week of recordings on healthy mice (Fig 5) which consists of the first week of recordings of the control group and the pre-stroke week in the robot group (see S2 Fig).

When looking at all events together, differences can be appreciated for the duration (p = 0.003) and the smoothness (p = 0.007) even without further splitting the events into specific types. In particular, the duration increases (Fig 5A) and the smoothness decreases (Fig 5C) during the subacute phase. The angles distribution for the subacute stroke group exhibits

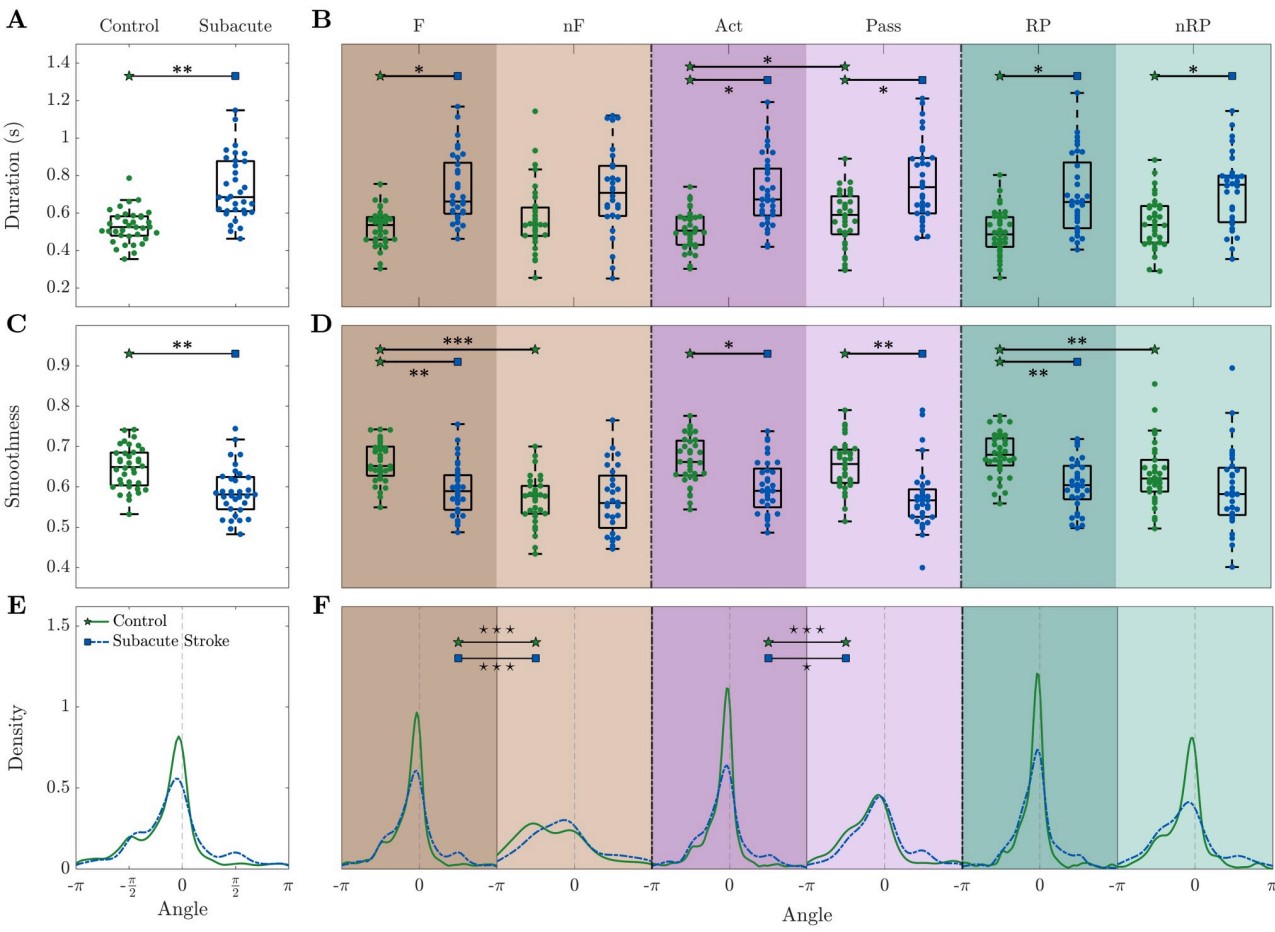

**Fig 5. The subacute phase (one week after stroke) is characterized by an increase of the event duration and decrease of smoothness.** During the subacute phase, **(A-B)** the event duration is increased, **(C-D)** smoothness is decreased, and **(E-F)** the direction of propagation is more spread.—Results refer to the first week of recording after stroke. Duration and angle are weighted by smoothness. Markers in (A-D) refer to the average value per day. In (A,C) each point is the average of all events per day per mouse, while in (B,D) each point is the average of all events of a specific type per day per mouse. Within each box in (A-D), the central mark indicates the median, and the bottom and top edges of the box indicate the $25^{th}$ and $75^{th}$ percentiles, respectively. Control n = 8 mice, Subacute Stroke n = 8 mice. See S2 Table for p-values of statistical tests, "⋆" refers to difference in variance and "∗" refers to difference in mean.

a flatter distribution and two secondary peaks in $-\pi/2$ and $\pi/2$, indicating the presence of a more heterogeneous pool of events (Fig 5E).

When splitting the results into event types (Fig 5B, 5D and 5F), a common tendency for duration, smoothness and angle is that the stroke condition attenuates differences of the indicators between event types. For both duration and smoothness (Fig 5B and 5D), while the control group presents significant variations with respect to the type of event (Act-Pass p = 0.04 for duration, and F-nF p = $10^{-10}$ and RP-nRP p = 0.008 for smoothness), the subacute stroke group is characterized by smaller fluctuations in the mean value (Act-Pass p = 0.059 for duration, and F-nF p = 0.082 and RP-nRP p = 0.51 for smoothness). For the duration (Fig 5B), significant differences in mean between healthy and stroke mice are manifested for force (F, p = 0.026), active (Act, p = 0.034), passive (Pass, p = 0.035), reward pulling (RP, p = 0.014), and not reward pulling (nRP, p = 0.028) events. Regarding the smoothness (Fig 5D), significant differences can be found in the same event types as for duration, non rewarded pulls (nRPs) excluded (p = 0.004, 0.01, 0.002, 0.005). For the angle (Fig 5F), the dissimilarities in the

control group between event types are preserved in the subacute stroke condition (F-nF and Act-Pass, control $p < 0.001$ and robot $p < 0.05$).

Altogether, our three spatiotemporal propagation indicators are able to distinguish between healthy and stroke mice. The subacute phase after stroke leads to more heterogeneous events characterized by longer duration, lower smoothness, and flattened distributions of the angle. Moreover, in contrast to the healthy condition, for the subacute stroke condition cortical propagation features are not able to discriminate event types.

## Combined rehabilitative treatment induces short duration and high smoothness of cortical propagation patterns

Our previous results comparing motor, pharmacological and combined therapy after focal stroke demonstrated that only a rehabilitation protocol coupling motor training with reversible inactivation of the contralesional cortex was able to promote recovery in forelimb functionality [24]. Here we first analyzed the propagation patterns in spontaneously recovered mice (untreated stroke group) one month after the lesion (S3 Fig). Results show that untreated animals present only minor modulations in the propagation features with respect to the subacute phase. The only statistical difference between the two groups is found for smoothness, which is lower for the untreated group, maintaining the same trend induced by the stroke in the subacute phase (compare Fig 5).

We hypothesized that rehabilitative treatments alter the spatiotemporal propagation patterns, and especially reverse the trend observed in the subacute stroke phase. In particular, given the fact that the combined treatment is the only treatment that leads to a generalized recovery (see S4 Fig), we wondered if we can find for this treatment spatiotemporal features of cortical propagation that reflect this unique success. We thus compared the spatiotemporal propagation indicators in treated animals with spontaneously recovering mice (untreated stroke group). In detail, we evaluated the consequences on the propagation patterns in mice treated with motor training alone (robot group), pharmacological silencing of the homotopic cortex alone (toxin group) or a combination of both (combined treatment group), see Fig 6. Note that results in Fig 6 refer to the fourth week after stroke in order to consistently compare the groups (since they all have been recorded during the fourth week) and because we are interested in the outcome of treatments.

Indeed, the most striking result emerging from this analysis is that the combined treatment group is greatly separated from all the other groups. When looking at all events together (Fig 6A), the events of the combined treatment group display a shorter duration compared to untreated ($p = 10^{-4}$), robot ($p = 10^{-5}$), and toxin ($p = 0.006$). The biggest differences can be observed for combined treatment versus untreated and robot groups; they are significant not only for all events together but also for each event type separately (all $p < 0.006$ and all $p < 10^{-4}$, respectively), see Fig 6B.

Among all the characteristics investigated, the marker that distinguishes most clearly between the combined treatment group and all the others is smoothness. The events of the combined treatment group display a greater smoothness compared to the untreated ($p = 10^{-5}$), robot ($p = 10^{-4}$), toxin ($p = 0.003$) groups (Fig 6C). Again, this statement applies not only to all events together, but also to each event type individually (all $p < 10^{-4}$ for untreated and robot, all $p < 0.023$ for toxin), see Fig 6D.

Dissimilarities in the angle distributions are able to capture the difference between the robot group and both untreated and toxin groups (Fig 6E and 6F). Note that all three indicators of the combined treatment group appear to be consistent when distinguishing different

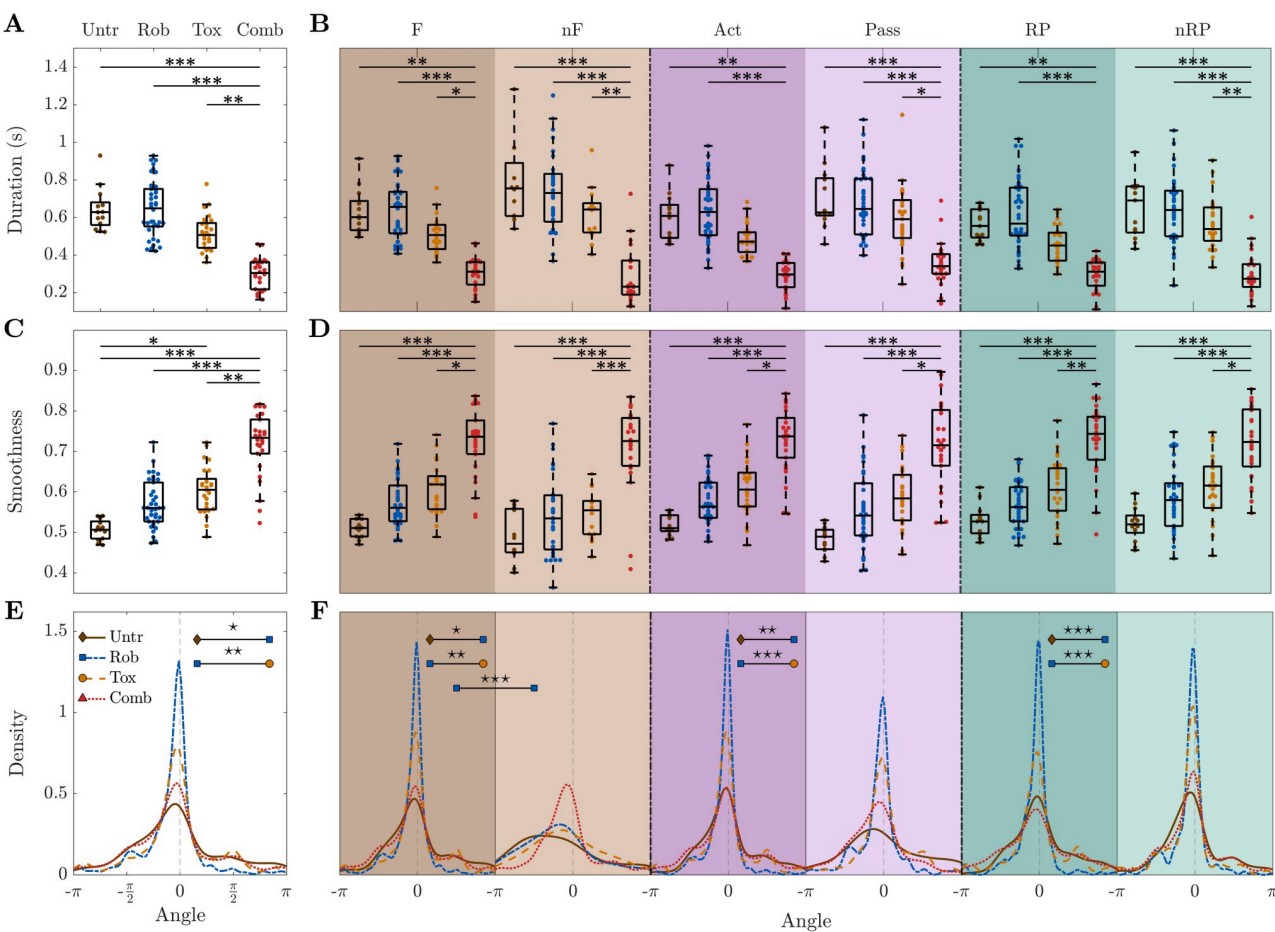

**Fig 6. Combined treatment group is characterized by higher smoothness and shorter duration than any other group.** For all types of event, combined treatment group events are the shortest **(A-B)** and their smoothness is the highest **(C-D)**. For the combined treatment group the distribution of the angles does not vary depending on the type of event **(E-F)**.—Results refer to the fourth week of recording after stroke. Duration and angle are weighted by smoothness. Markers in (A-D) refer to the average value per day. In (A,C) each point is the average of all events per day per mouse, while in (B,D) each point is the average of all events of a specific type per day per mouse. Within each box in (A-D), the central mark indicates the median, and the bottom and top edges of the box indicate the 25th and 75th percentiles, respectively. Untreated stroke group n = 4 mice, robot group n = 8 mice, toxin group n = 5 mice, combined treatment group n = 6 mice. For P-values of statistical tests see S3 Table, "★" refers to difference in variance and "*" refers to difference in mean.

event types, meaning force, non-force, active, passive, reward and non-reward pulling events display similar average values. The complete list of p-values can be found in S3 Table.

Additionally, we evaluated if propagation features were restored to pre-stroke levels or if the treated condition ended in a new state (S5 and S6 Figs). We analyzed longitudinal data starting from the second week of recording up to one month after stroke, by comparing motor treated (robot and combined) with healthy mice (control group), see S5 Fig. Results show that propagation features averaged over the training weeks were partially restored to pre-stroke levels for the robot group (no statistical differences were found with the control group), while the combined treatment group ended in a new state with very different values from both pre-stroke and motor-treated conditions. In details, the events of the combined treatment group display shorter duration ($p = 10^{-4}$, $p = 10^{-6}$) and a greater smoothness ($p = 0.025$, $p = 10^{-4}$) compared to the control and robot groups. For a complete list of p-values refer to S7 Table. When looking at the temporal evolution of the recovery (S6 Fig), in the weeks that follow the

stroke the robot group shows a significant variation: the smoothness is lower and the duration is longer. This variation decreases over time in both cases. While the duration reaches again values comparable to healthy mice already after the second week of training, the difference in smoothness seems to oscillate without stabilizing. Interestingly, the rehab group presents a behavior qualitatively comparable to the control group, i.e., a stable trend, but with very different values.

In summary, the combined treatment group is significantly different from all the other groups. In particular, it is characterized by the shortest duration and the highest smoothness. These differences can not only be observed for all events together, but also for all event types (force, non-force, active, passive, reward pulling, non-reward pulling) separately. Once more, this confirms that the combined therapy, the only one leading to generalized recovery, is associated with a state different from pre-stroke conditions which shows new and unique propagation features.

## Discussion and conclusions

In this study we employed an improved version of our recently proposed SPIKE-order analysis [23] to sequences of wide-field fluorescence calcium images from the dorsal cortex of awake behaving mice. We defined three propagation indicators that characterize the duration, the angle of propagation and the smoothness of movement-evoked global events. This new way of quantifying variations in the spatiotemporal propagation patterns allowed us to track damage and functional recovery following stroke.

We found that in healthy mice all three indicators of spatiotemporal propagation display a very high degree of consistency over time. For animals with subacute stroke the propagation patterns of the global events through the injured hemisphere are altered. The most prominent consequence is a large increase in global event duration and a decrease in smoothness over the ipsilesional hemisphere. We compared spontaneous recovery with three different rehabilitation therapies, motor training, transient pharmacological silencing of the homotopic cortex and a combination of both. While all of these treatments have an impact on the spatiotemporal propagation patterns, the combined therapy, promoting a generalized recovery of forelimb functionality, leads to an increased propagation efficacy, different from pre-stroke conditions, with very fast and smooth patterns.

### Comparison with existing methods

Most analysis tools for wide-field optical images perform simple pairwise correlation and time lag analysis (Pearson correlation and phase synchrony), which are window- and not event-based as in this study (e.g. [32]). Functional connectivity techniques typically result in a graph or network describing the relationship between different brain regions [33, 34]. Directionality is explored in [35] by using Granger causality, which is dependent on a priori selection of regions of interest (ROIs). Since these commonly used analyses tools are based on averaged activity under resting state, important information on single events is missing. This is especially important when considering motor evoked activity, which is directly associated with the execution of single movements (active forelimb pulling in our case). The detailed propagation analysis we showed here would not be possible with the widely used optical flow techniques [36, 37] which instead focus on velocity vector fields and their complex patterns (e.g. sources and sinks) but do not deal explicitly with temporal order. Our methods used here contain some similarities but also crucial differences with a very recent analysis performed on cortical slow waves in anesthetized mice [38]. On the one hand, both algorithms apply exactly the same criteria of unicity and globality in the identification of the spatiotemporal patterns. On

the other hand, the core of our global event detection is automated spike matching (via adaptive coincidence detection) while the wavehunt pipeline relies on an iterative procedure to cut the time series into distinctive waves. Apart from the different types of data, the two studies are also complementary in scope: while the focus of [38] lies on the excitability of the neuronal population, the dominant origin points and the velocity of the slow waves, we perform a thorough investigation of the spatial propagation pattern of each global event.

As a final remark on the methods, we would like to stress that the approach used here is universal and could easily be adapted to functional techniques (including electroencephalography and functional Magnetic Resonance Imaging) in many other clinical settings. For example, it could be extended to disorders of the central nervous system similarly associated with alterations in the spatiotemporal propagation of brain activity, from traumatic brain injury to autism.

## Cortical propagation features in healthy mice

We first tested the discrimination capabilities of this approach for different classes of behavioral events and used it to characterize global events over most of the dorsal cortex. Results show that under our experimental paradigm, in all conditions, global events are occurring predominantly when the mouse is actively applying force during either active retraction or passive extension of the affected forelimb. During glogal events, we observe a large wave traveling across most of the imaged cortex, where a wide population of excitatory neurons is progressively depolarized. Previous studies suggested that wave propagation is relevant to cortical information transfer during movement preparation and execution [39, 40]. Traveling waves could contribute to cortical processing by determining when and where the cortex is depolarized, in relation to a motor event [41]. Also, traveling waves in the motor cortex have been suggested to facilitate sequences of activation in muscle representations in preparation of goal-directed movements [42]. In our study, each of the novel spatiotemporal features of motor-evoked calcium wave propagation we described could encode for specific aspects of the forelimb movement. In line with this hypothesis, angle, duration and smoothness of the global events change with behavioral event type (e.g. if the event is associated with the application of force or not) in healthy subjects. This finding is in line with a recent study showing different propagation patterns across the cortex for mice engaged in a visual task depending on the type of the behavioral event (active vs passive, hit vs misses, ipsilesional vs controlesional) [43].

The drastic change observed in the angle distribution of global events between force and non-force events implies that activity propagates from medial to lateral regions. This is in accordance with previous findings based on space-frequency single value decomposition analysis showing that at the naive stage, the activity propagated from retrosplenial cortex in a radial direction [20]. The mediolateral propagation of the global events suggests the progressive involvement of the retrosplenial cortex during the exertion of the reward pull. Indeed, it has been previously reported that retrosplenial cortex is more correlated with sensory cortices during locomotion vs quiescence [44] suggesting the presence of a network switch to allow the processing of sensory information during locomotion. In this view, the higher accumulation of angles at 0 degrees observed when comparing force vs non-force events could represent the hallmark of such network switch. Interestingly, a similar propagation pattern has been observed applying optical flow analysis to calcium imaging over the same cortical areas when windowing cortical activity around hippocampal sharp wave ripples during sleep [45] and corresponds to one of the two major propagation patterns observed during slow wave sleep [46]. Our findings extend these results to the awake condition during motor execution.

## Subacute phase after stroke

Stroke strongly affects the spatial propagation within the cortex during the execution of a pulling task. The analysis of spatiotemporal propagation patterns evoked by stimulation or voluntary movements represents a fundamental means to investigate functional remapping in order to better understand post stroke reorganization.

In our study, subacute stroke is characterized by less coherent direction of the propagation, lower smoothness, and longer duration than healthy animals. Also, cortical propagation properties are very heterogeneous across global events and the differences between behavioral event types are lost. Wave-like propagation (and possibly information transfer) during global events is likely impaired, as indicated by the higher duration of the motor-evoked cortical calcium wave and lower smoothness. Together with our previous findings showing reduced localization of motor-evoked activation in the untreated stroke [6], these data suggest a complex dysregulation of information processing and integration across the cortex that possibly give rise to a fragmented motor command. In a previous work Murphy and collaborators [47], by applying intrinsic optical signal and fluorescence imaging, described modifications in spatiotemporal propagation elicited by sensory forelimb stimulation both in subacute and chronic phase after stroke in the forelimb cortex. In agreement with Brown et al., who observed during the subacute phase an increase of time to peak cortical signal evoked by sensory stimulation, our results revealed an increment of duration of motor-evoked cortical response, showing a delayed activation of cortical regions neighboring the stroke core due to the damage. The comparison between these results reveals that though applying opposite approaches (i.e. bottom-up for sensory stimulation and top-down for motor task execution) a similar cortical response was observed. Moreover, an fMRI study in the subacute phase by Dijkhuizen and colleagues [48] showed that the stimulation of the unpaired forelimb induces a small response detected in the ipsilesional hemisphere in M1 and sFL cortex and in more distal regions both in rostral and caudal direction. A similar observation was made by Harrison and collaborators [14] revealing that motor maps were more diffuse after motor-targeted stroke during sensory stimulation, with a decrease in correlation between neighboring pixels. The diffuse activation in response to forelimb stimulation observed in those previous works is in agreement with our results that reveal the absence of a clear pattern of cortical propagation, as highlighted by low smoothness, in the subacute phase after stroke.

## Comparison of different rehabilitation paradigms

In the chronic phase after stroke, the establishment of a new pattern in the global event could be pivotal in gaining an optimized computation and communication between subsystems of the brain [49]. Results show that task-specific spatiotemporal patterns of neuronal activity spreading across the cortex can indeed be modulated, and how this modulation occurs is dependent on the therapeutic approach.

Different rehabilitative paradigms result in different rehabilitative outcomes; in particular, not all treatments promote generalized recovery of forelimb functionality. In this study, we purposely selected three rehabilitative treatments (robot, toxin, and combined treatment) that all induced changes in the neural signals but only the combined treatment leads to generalized recovery. Combined rehabilitation profoundly altered the propagation of global events as compared to both healthy (control) and post-stroke single treated (toxin and robot) mice. The differences in the propagation features induced by stroke in the subacute phase decline during weeks of motor training in the robot group, in fact already by the second week of training duration reaches values comparable to healthy mice. Also the smoothness was on average comparable to healthy mice after robotic training. Our findings on the chronic phase of the robot

group are in agreement with what we observed in [50] where repetitive motor training induced a task-dependent spatial segregation similar to healthy mice though unaccompanied by functional recovery [24]. Each treatment had an impact on separate propagation features. In fact, while during the chronic phase only robot mice showed significant differences in the direction of cortical propagation (angle) compared to untreated stroke mice, the toxin group had peculiar (and significantly different) features in smoothness. Combination of the two treatments results in profound global changes in all indicators, with new features completely different from all the other groups. More in detail, combined treatment mice show a decrease of duration and greater smoothness with respect to control and robot mice, indicating the emergence of faster and more directed patterns of propagation. Such temporally compressed and reliable cortical activity sequences may be associated with a more effective trigger of subcortical movement machinery [20].

In addition, the substantial increase in smoothness after combined rehabilitation finds a nice correlate in the segregation of motor representation illustrated in preclinical [6] and clinical studies [51]. In these works, improved motor functionality induced by post-stroke combined rehabilitation is associated with a more focused brain activation during the execution of a motor task [50]. Importantly, generalized recovery of forelimb functionality in combined treatment mice (see [6, 24]) is not necessarily associated with recovery of pre-stroke spatiotemporal propagation features. Indeed, the results on all motor-evoked spatiotemporal propagation indicators suggest that the combination of contralesional inactivation and motor training acts towards the establishment of new propagation patterns rather than the restoration of pre-stroke features. The combined rehabilitation may indeed entrain the cortical network on a new spatiotemporal pattern for the global events, where the duration of the traveling wave is shorter and smoother. In a previous study we showed that also rostro-caudal propagation is faster, and this is associated with a higher correlation in the activity of spared motor-associated areas in REHAB mice [6]. Taken together, these results suggest that a new communication scheme for motor control emerges in REHAB mice, where information transfer is more efficient (small duration) and coherent (increased smoothness), possibly suggesting an improved effective and structural connectivity between distal regions. Future correlative studies could help disentangle the relationship between altered propagation and anatomical connectivity [52].

Although the functional outcome after stroke can be predicted on the basis of motor functions tests with or without neurological or neuroimaging assays [2, 53], here we test the idea if cortical propagation measures are affected not only by stroke but also by different therapies. In these regards, our study differs from previous ones that have established that the degree of recovery after stroke can be predicted from a baseline assessment of stroke injury, neural function and clinical status [2, 54–56]. Here, instead, we are not making predictions on the recovery based on acute stroke assessment, here we are showing that the time course of the cortical propagation indicators after stroke changes upon different therapies or spontaneous recovery and therefore these measures can be employed as a proxy to assess and monitor recovery. Finally, additional studies are required to check if our propagation measure can be used to predict functional outcome and how they compare to other measures that predict functional outcome in the stroke acute phase.

In summary, our detailed spatiotemporal analysis of global activation patterns during longitudinal motor training provides a powerful tool to quantify the success of different state-of-the-art rehabilitation paradigms. Propagation-based biomarkers may deliver new and unforeseen information about the brain mechanisms underlying motor recovery which could pave the way towards a much more targeted post-stroke therapy.

## Source codes

The main method used here, SPIKE-Order, is implemented in three free and publicly available software packages. Results in this study were obtained using cSPIKE (http://wwwold.fi.isc.cnr.it/users/thomas.kreuz/Source-Code/cSPIKE.html) (Matlab command line with MEX-files). A Matlab-based graphical user interface, SPIKY (http://wwwold.fi.isc.cnr.it/users/thomas.kreuz/Source-Code/SPIKY.html) [30], and a Python library, PySpike (http://www.pyspike.de) [57], are available as well.

## Supporting information

**S1 Fig. Standard atlas of brain regions and their acronyms.**
(EPS)

**S2 Fig. As expected, the pre-stroke condition shows the same behavior as the control group for all three propagation indicators. (A-B)** duration, **(C-D)** smoothness, and **(E-F)** angle.—Duration and angle are weighted by smoothness. Markers in (A-D) refer to the average value per day. Within each box in (A-D), the central mark indicates the median, and the bottom and top edges of the box indicate the $25^{th}$ and $75^{th}$ percentiles, respectively. Control group n = 3 mice, pre-stroke group n = 5 mice. P-values of statistical tests in S4 Table.
(EPS)

**S3 Fig. Compared to the subacute phase, spontaneously recovered mice (untreated stroke group) present only minor modulations in the propagation indicators. (A-B)** duration, **(C-D)** smoothness, and **(E-F)** angle.—Duration and angle are weighted by smoothness. Markers in (A-D) refer to the average value per day. Within each box in (A-D), the central mark indicates the median, and the bottom and top edges of the box indicate the $25^{th}$ and $75^{th}$ percentiles, respectively. Control group n = 3 mice, pre-stroke group n = 5 mice. P-values of statistical tests in S5 Table.
(EPS)

**S4 Fig. Combined treatment leads to generalized recovery.** Asymmetry in the spontaneous forelimb use measured by the Schallert cylinder test once per week and rescaled by the pre-stroke (day -5) value for each mouse. Significant differences are calculated for the last day of recording (day 30). P-values of statistical tests in S6 Table.
(EPS)

**S5 Fig. Combined treatment group is characterized by shorter duration and higher smoothness. (A-B)** For all types of event, combined treatment group events are the shortest. **(C-D)** For all types of event, the smoothness of the combined treatment group is higher than for the robot group. **(E-F)** For the combined treatment group the distribution of the angles does not vary depending on the type of event.—Longitudinal data starting from the second week of recording up to one month after stroke. Duration and angle are weighted by smoothness. Markers in (A-D) refer to the average value per day. Within each box in (A-D), the central mark indicates the median, and the bottom and top edges of the box indicate the $25^{th}$ and $75^{th}$ percentiles, respectively. Control group n = 3 mice, robot group n = 8 mice, combined treatment group n = 6 mice. P-values of statistical tests in S7 Table, "⋆" refers to difference in variance and "∗" refers to difference in mean.
(EPS)

**S6 Fig. Temporal evolution of duration (A) and smoothness (B) for all groups.** In both cases, the difference between control and subacute stroke groups is considerably larger for the first three days. Moreover, the difference in the event duration tends to diminish over days.

Event duration is always shorter for the combined treatment group, smoothness is always higher. The control group shows a consistent trend, suggesting that motor training alone has no effect on either duration or smoothness, as already described in the previous sections for healthy mice. After the stroke, the robot group shows a significant variation: the duration is longer and the smoothness is lower; in both cases this variation decreases over time. While the duration reaches again values comparable to healthy mice already after the second week of training, the difference in smoothness seems to oscillate without stabilizing.
(EPS)

**S1 Video. Movie depicting the propagation of activity during the training cycle shown in Fig 1C.** As the vertical green line superimposed over subplots (A) to (D) moves forward in time (from -0.12 seconds before to 1.28 seconds after the threshold crossing of the force), in the Calcium image in subplot (E) a propagation of activity from left to right can be seen. This is in accordance with the propagation matrix shown in subplot (F) which color-codes the temporal order of activation from leader (red) to follower (blue). (A) Status of the robotic sled. (B) Force applied by the mouse during the retraction movement. (C) Average calcium signal over all pixels. (D) Raster plot obtained from the threshold crossings of individual pixels versus time. (E) Calcium imaging sequence of cortical activation, superimposed on the standard atlas of brain regions. (F) The propagation matrix is obtained by projecting the relative order of these threshold crossings onto the 2D-recording plane.
(AVI)

**S1 Appendix. Materials and methods. Experimental design**: Mice; Surgical Procedures; Motor Training Protocol on the M-Platform; Wide-Field Fluorescence Microscopy; Schallert cylinder test. **Signal processing and data analysis**: Preprocessing; Event detection; SPIKE-Order; Categorization of events; Three propagation indicators: Duration, Angle, Smoothness. Statistical Tests.
(PDF)

**S2 Appendix. SPIKE-Order method.** Adaptive Coincidence Detection; SPIKE-Synchronization; SPIKE-Order; Synfire Indicator.
(PDF)

**S1 Table. P-values for Fig 4.**
(PDF)

**S2 Table. P-values for Fig 5.**
(PDF)

**S3 Table. P-values for Fig 6.**
(PDF)

**S4 Table. P-values for S2 Fig.**
(PDF)

**S5 Table. P-values for S3 Fig.**
(PDF)

**S6 Table. P-values for S4 Fig.**
(PDF)

**S7 Table. P-values for S5 Fig.**
(PDF)

## Acknowledgments

We thank Shih-Chieh Lin and Cristina Spalletti for their reading of the manuscript and their insightful comments.

## Author Contributions

**Conceptualization:** Gloria Cecchini, Alessandro Scaglione, Anna Letizia Allegra Mascaro, Thomas Kreuz.

**Data curation:** Alessandro Scaglione, Emilia Conti.

**Formal analysis:** Gloria Cecchini, Curzio Checcucci, Thomas Kreuz.

**Funding acquisition:** Francesco Saverio Pavone.

**Investigation:** Gloria Cecchini, Alessandro Scaglione, Anna Letizia Allegra Mascaro, Curzio Checcucci, Emilia Conti, Duccio Fanelli, Thomas Kreuz.

**Methodology:** Gloria Cecchini, Alessandro Scaglione, Anna Letizia Allegra Mascaro, Duccio Fanelli, Thomas Kreuz.

**Project administration:** Francesco Saverio Pavone.

**Resources:** Francesco Saverio Pavone.

**Software:** Gloria Cecchini, Thomas Kreuz.

**Supervision:** Duccio Fanelli, Francesco Saverio Pavone, Thomas Kreuz.

**Validation:** Curzio Checcucci.

**Visualization:** Gloria Cecchini, Thomas Kreuz.

**Writing – original draft:** Gloria Cecchini, Alessandro Scaglione, Anna Letizia Allegra Mascaro, Curzio Checcucci, Emilia Conti, Thomas Kreuz.

**Writing – review & editing:** Gloria Cecchini, Alessandro Scaglione, Anna Letizia Allegra Mascaro, Curzio Checcucci, Emilia Conti, Ihusan Adam, Duccio Fanelli, Roberto Livi, Thomas Kreuz.

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
