## [Decision Letter · Decision Letter 0]

15 Feb 2021

Dear Dr. Cecchini,

Thank you very much for submitting your manuscript "Cortical propagation as a biomarker for recovery after stroke" for consideration at PLOS Computational Biology.

As with all papers reviewed by the journal, your manuscript was reviewed by members of the editorial board and by several independent reviewers. In light of the reviews (below this email), we would like to invite the resubmission of a significantly-revised version that takes into account the reviewers' comments.

We cannot make any decision about publication until we have seen the revised manuscript and your response to the reviewers' comments. Your revised manuscript is also likely to be sent to reviewers for further evaluation.

Sincerely,

Arvind Kumar, Ph.D.

Guest Editor

PLOS Computational Biology

Kim Blackwell

Deputy Editor

PLOS Computational Biology

Reviewer's Responses to Questions

**Comments to the Authors:**

Reviewer #1: In this paper, the authors applied an original and advanced analysis approach to measure the effect of stroke and post-stroke rehabilitation paradigms on spatiotemporal activation patterns over the cortex in awake mice. Activation patterns were measured with wide-field calcium imaging during forelimb movements in a robotic system. Specific parameters that describe propagation patterns, i.e. duration, smoothness and angle, were shown to be altered after stroke. Cortical activity propagation after stroke was slowed down and more irregular. Combined treatment led to renewed patterns, which were even faster and smoother than before stroke. These results provide new insights into neural recovery processes, although the biological underpinnings remain to be elucidated.

Specific comments:

1) The authors claim in the Introduction that functional outcome after stroke is hardly predictable. However, several studies have shown that motor outcome can be well predicted from the degree of motor impairment in the acute stage. This can be assessed with motor functions tests, possibly supplemented with measurement of neurophysiological or neuroimaging biomarkers. These standard methods of monitoring and prediction of motor outcome should be mentioned and discussed in comparison to the authors’ approach of tracking recovery from cortical activity patterns.

2) It is not (directly) clear from the Introduction that the study involved experimental stroke studies in mice. Please adjust.

3) Measurements were started in the first week after stroke. This is the subacute post-stroke stage in mice, and not the acute stage. Please correct.

4) Sample sizes of the experimental groups varied between 3 and 8. Please explain why these were not equal.

5) Fig. 1b shows different color-coded symbols. What does the number of the symbols (for example the thirteen brown diamonds for ‘Untr’ at week 4) represent? Please clarify in the figure legend. If animals were measured multiple times, then I recommend to also code for individual animals in the figure.

6) It is unclear it the lesion area was included in the imaging window. The lesion contains functionally impaired, structurally damaged or lost neurons, which would significantly affect the cortical activity and propagation profiles. Moreover, the extent of the lesion may have differed between time-points and between groups. How did the authors deal with this issue?

7) Unfortunately, there was no link to the video (S1 Video).

8) In the Discussion the authors relate their findings to similar results from previous studies. However, discussion of the possible biological underpinnings of disturbed propagation of cortical activity after stroke, and possible mechanisms of recovery in response to rehabilitation treatment, is lacking. The authors should at least speculate on possible biological mechanisms, for example related to changes in neuronal connectivity and transmission, that could explain their findings.

Reviewer #2: This study induced stroke in transgenic mice and used calcium imaging to study cortical activity propagation patterns and their relation to motor function and recovery. It applies a model of spike train dynamical similarity to calcium activity and uses analyses of the outcome of this mode to explain differences in motor performance.

The idea that different patterns of spatiotemporal dynamics could underlie motor recovery and function is important, and the paper asks this question in an interesting way. The data are high quality. And I am especially impressed with the attempt to apply measures of spreading activity that have more complex dynamics than simple time adjacency of activity in different areas (standard “functional connectivity,” FC). The idea of tracking where and how a neural “message” propagates is an intriguing and attractive one. Despite the limits of resolution in calcium imaging, which give us only a gross picture of how a motor “message” travels, I think any new way to trace signals over more than one synapse or population in the awake brain is useful. However, I think the authors should do more to justify the particular dynamical model used here, which centers on signal duration, smoothness, and direction of travel.

First, SPIKE is a somewhat misleading name. I realize there is a history to this usage but it seems inappropriate here. “Events” in this context are misleadingly referred to, following the model’s moniker, as “spikes.” The time resolution in the present study is 1000x lower compared to the recordings for which the method was originally developed (Satuvuori et al., 2017). Maybe call calcium events “pulses” instead (or similar).

However, this is not just a matter of semantics: no evidence is presented suggesting that dynamical spreading serves or obeys the same functional principles at the millisecond scale as at the second scale. I rather suspect they follow different constraints and serve different goals. Certainly, the SPIKE model is a welcome attempt to move past simplistic FC models but I think this needs to be justified more. The model also introduces three new parameters to augment standard FC analysis, along with numerous pre-processing steps. This is the innovation in the paper, but it also a priori makes it more likely that the model will discover differences in the functional data. If the model itself exploited some compelling aspect of the dynamic organization of neural activity (e.g., communication efficiency over the network, etc.), this would be fine. But at present, the model’s application is not well justified. Thus, it’s hard to say whether the model succeeds because it captures something important about dynamical activity leading to successful motor achievement or because it has been tuned to the data via the addition of parameters and/or because of the winnowing of the data in pre-processing.

In any case, the ability to distinguish functional outcomes using the present method is hard to discern without attempts at prediction. Differences in the dynamical measures are small and there is a lot of overlap in the distributions for these measures. One could do ridge regressions or similar on individual sessions to see if its group membership can be predicted based on smoothness, duration etc. Without this, I don’t think we can really say these measures constitute a “biomarker.” I would say they are more like average trends or apparent distinctions in dynamics.

Minor Points:

I’m not sure “smoothness” is the appropriate term for the measure employed here. It seems more like the linearity or maybe the dimensionality of the data. What looks like a dominant pattern of propagation to our eyes—or to an orthogonal transform like SVD—does not necessarily constitute the “smooth” spreading of activity.

How big are the calcium signals? Fig 2 gives arbitrary units but these should instead be given as Z-scores.

Forgive me if I have misread Fig 5 but the average data (leftmost column) for the controls looks possibly inconsistent with the other columns. I assume the leftmost column is an average across conditions (F/nF, Act/Pass, RP/nRP). If so, I am not certain that the variance could have gone down quite so much compared to the individual conditions. Is it possible that the vertical scales are not the same? Of course, it’s hard to tell by eye.

Line 502: I would not call calcium imaging “non-invasive.”

**Have all data underlying the figures and results presented in the manuscript been provided?**

Reviewer #1: Yes

Reviewer #2: Yes

PLOS authors have the option to publish the peer review history of their article (what does this mean?). If published, this will include your full peer review and any attached files.

Reviewer #1: No

Reviewer #2: **Yes: **Daniel Graham
---

## [Decision Letter · Decision Letter 1]

13 Apr 2021

Dear Dr. Cecchini,

We are pleased to inform you that your manuscript 'Cortical propagation tracks functional recovery after stroke' has been provisionally accepted for publication in PLOS Computational Biology.

Best regards,

Arvind Kumar, Ph.D.

Guest Editor

PLOS Computational Biology

Kim Blackwell

Deputy Editor

PLOS Computational Biology

Reviewer's Responses to Questions

**Comments to the Authors:**

Reviewer #1: The authors have adequately responded to my comments.

Reviewer #2: The authors have addressed my comments well. I congratulate them on a very nice paper.

**Have all data underlying the figures and results presented in the manuscript been provided?**

Reviewer #2: Yes

PLOS authors have the option to publish the peer review history of their article (what does this mean?). If published, this will include your full peer review and any attached files.

Reviewer #1: **Yes: **Rick M. Dijkhuizen

Reviewer #2: **Yes: **Daniel Graham

**Have the authors made all data and (if applicable) computational code underlying the findings in their manuscript fully available?**

Reviewer #1: Yes

---

## [Editor Report · Acceptance letter]

6 May 2021

PCOMPBIOL-D-20-02241R1 

Cortical propagation tracks functional recovery after stroke

Dear Dr Cecchini,

I am pleased to inform you that your manuscript has been formally accepted for publication in PLOS Computational Biology. Your manuscript is now with our production department and you will be notified of the publication date in due course.

With kind regards,

Katalin Szabo
